# Molecular and Hormonal Regulation of Leaf Morphogenesis in Arabidopsis

**DOI:** 10.3390/ijms21145132

**Published:** 2020-07-20

**Authors:** Shahid Ali, Naeem Khan, Linan Xie

**Affiliations:** 1College of Life Sciences, Northeast Forestry University, Harbin 150040, China; 2Department of Agronomy, Institute of Food and Agricultural Sciences, University of Florida, Gainesville, FL 32611, USA; naeemkhan@ufl.edu; 3Key Laboratory of Saline-Alkali Vegetative Ecology Restoration, Ministry of Education, College of Life Science, Northeast Forestry University, Harbin 150040, China

**Keywords:** adaxial/abaxial, leaf development, miRNAs, shoot apical meristem, transcription factors

## Abstract

Shoot apical meristems (SAM) are tissues that function as a site of continuous organogenesis, which indicates that a small pool of pluripotent stem cells replenishes into lateral organs. The coordination of intercellular and intracellular networks is essential for maintaining SAM structure and size and also leads to patterning and formation of lateral organs. Leaves initiate from the flanks of SAM and then develop into a flattened structure with variable sizes and forms. This process is mainly regulated by the transcriptional regulators and mechanical properties that modulate leaf development. Leaf initiation along with proper orientation is necessary for photosynthesis and thus vital for plant survival. Leaf development is controlled by different components such as hormones, transcription factors, miRNAs, small peptides, and epigenetic marks. Moreover, the adaxial/abaxial cell fate, lamina growth, and shape of margins are determined by certain regulatory mechanisms. The over-expression and repression of various factors responsible for leaf initiation, development, and shape have been previously studied in several mutants. However, in this review, we collectively discuss how these factors modulate leaf development in the context of leaf initiation, polarity establishment, leaf flattening and shape.

## 1. Introduction

Leaves are the primary organs responsible for photosynthesis and photoperception, and play a key role in plant growth. Their development starts from the shoot apical meristem (SAM), which have a central zone (CZ) that houses pluripotent cells, and a peripheral zone (PZ), responsible for the leaf initiation and their development into a flattened structure [1]. In most plants, the leaf functions as a solar panel, where photosynthesis converts carbon dioxide and water into carbohydrates and oxygen [2]. Leaves are an excellent example of learning how complex organs arise from a simple structure. All leaves are initiated by the recruitment of cells flanking the SAM as simple rod-like primordia, later on, to get their final shape. There are three principal axes in a leaf, along which intrinsic genetic programs control leaf cell division, differentiation, and expansion. However, leaf morphogenesis is strictly controlled not only by intrinsic genetic factors but also by hormonal factors. Numerous series of events demonstrate that plant hormones, mostly small and simple molecules, play crucial roles in plant growth and development [3]. It was hypothesized how hormonal and genetic networks regulate leaf morphogenesis to enable the transformation of simple primordium into a complex organ with consistent shape and size, and to elaborate how these genetic networks generate plasticity in response to both endogenous and environmental signals. Since a deeper understanding of leaf development contributes to our overall comprehension of plant biology, this understanding can also be used to improve crop production. Therefore, it is important to unveil the molecular and hormonal regulation of leaf morphogenesis, including the initiation of leaf primordia, the determination of leaf axes, and the regulation of cell division in model plant *Arabidopsis thaliana*. The main focus of this review is to summarize the current knowledge and discuss the several fundamental aspects of leaf development, including a genetic regulatory framework that contributes to leaf initiation, leaf polarity determination, and leaf outgrowth and flattening. Furthermore, we also emphasize recent results that have strengthened our understanding of leaf development. The information obtained from the studies of *Arabidopsis thaliana* simple leaves will continue to provide basic knowledge about the formation and genetic mechanisms involved in compound leaves [4,5].

### 1.1. Maintenance of Shoot Apical Meristems (SAM) and the Leaf Initiation Gene Network

The stem cells of plant meristems generate new organs and tissues throughout the life of the plant. The above-ground organs in plants are formed through SAM, whereas below-ground organs are created by root apical meristems (RAM). Beside these two apical meristems, various other types of meristems exist in plants, such as lateral meristems (e.g., axillary in the node of the leaves and flowers), intercalary meristems (at the base of the monocots leaf blade), and transient stem-like meristemoids (the precursor for guard cells) [6,7]. The SAM is a dome-shaped structure that comprises a reservoir of stem cells, provides cells that form the branches, leaves, and flowers of the plant, and also retains its own identity. The SAM is organized into several distinct cell layers and various zones [8] (Figure 1). The development and maintenance of SAM are crucial for determining the spatiotemporal arrangement (e.g., clockwise, anticlockwise, spiral and whorled) of aerial organs around the stem. The process of arranging different organs spatially is known as phyllotaxis, which is species and stage-dependent [9,10,11].

In *Arabidopsis thaliana*, the SAM possesses three cell layers (L1–L3). An external two-cell layer forms tunica, where the L1 overlies L2. This layer divides by anticlinal cell division and grows in a two-dimensional fashion [12]. An inner layer (L3) divides both periclinal and anticlinal cell divisions in a mostly random fashion, which is commonly called corpus. These three histogenic layers are responsible for producing different parts; the L1 layer divides and forms the entire shoot epidermis while the L2 layer produces the photosynthesizing cells of the sub-epidermis. The L3 layer forms the internal tissue, pericycle, and other corpus cells [13].

The SAM layers are further subdivided into three functional domains or zones according to the function and division rate (Figure 1). The three zones include the central zone (CZ), peripheral zone (PZ), and rib zone (RZ), which is present below the CZ. The CZ is mainly responsible for the maintenance of SAM. The CZ contains both tunica and corpus cells in which the stem cells are present, and below the CZ is the organizing center (OC) [14]. The tunica and corpus cells of the CZ are symplasmically interconnected through the plasmodesmata [15,16]. Any alteration in the intercellular signals through the plasmodesmata affects normal growth and development [17].

The cells in the PZ divide at a faster rate to form leaves, branches, and floral parts than those in the CZ [15,18]. These functional and cytohistological zones are juxtaposed onto cell layers and govern the patterns of development [19,20]. The PZ cells are differentiated and have distinct identities that help to establish future organs such as leaf primordia [3,21]. Cells in the RZ develop into differentiating stems that support the SAM. The maintenance of SAM and the initiation of an organ through a specific phyllotaxis pattern requires strong signal coordination between the different factors such as hormones and genes [22,23].

The initiation and establishment of the shoot meristem in the embryonic stage of *Arabidopsis* relies on the expression of many different genes and certain signaling proteins. In plants, the NAC family, i.e., CUP-SHAPED COTYLEDON1 (CUC1), CUC2, and CUC3, are the major transcription factors required to initiate SAM and boundary formation (Table 1) [24,25,26,27]. The double mutant of *cuc1* and *cuc2* genes generates a fused cup-shaped cotyledon structure devoid of SAM. In *Arabidopsis*, for the development of SAM during embryogenesis and to sustain its function, the activity of class-1 *KNOTTED1-LIKE HOMEOBOX (KNOX*) gene *SHOOT MERISTEMLESS (STM)* is required, which is expressed throughout the SAM but downregulated in leaf developing cells [28,29]. In the organ (leaf) founder cells, the activation of organ-specific transcription factors and auxin accumulation inhibits the expression of the *KNOX* gene (Figure 1) [27]. Other members of the class-1 *KNOX* gene family, which are expressed in SAM, include *KNAT1/BREVIPEDICELLUS (BP*), *KNAT2*, and *KNAT6. STM* is a central regulator of SAM organization and development, and stronger alleles, such as the *stm-1* mutant, totally failed to establish SAM during embryogenesis [30]. In contrast and under normal conditions, other related genes, such as *KNAT1/BP*, *KNAT2*, and *KNAT6*, have no discernible effect on SAM. In the leaf primordium, the MYB domain transcription factors, ASYMMETRIC LEAVES1 (AS1) from *Arabidopsis*, ROUGH SHEATH2 (RS2) from maize and PHANTASTICA (PHAN) from *Antirrhinum* (these are collectively named *ARP* genes), repress the expression of the *KNOX1* gene (Table 1) [31,32]. Furthermore, to maintain the meristematic state of the stem cells, the *STM* inhibits the expression of *AS1* in the shoot apex [28,32]. *KNOX1* gene *BREVIPEDICELLUS (BP)* and *KNAT2* gene expression are directly repressed by a repressor complex containing AS1, AS2 [33,34]. STM induces the cytokinin (CK) biosynthesis gene, *isopentenyl transferase (AtIPT7)* which encodes for the enzyme that contributes to the production of active CK in the SAM layer L1; CK interacts with other systematic signals and controls the meristem size and functions [35].

The *WUSCHEL*-related homeobox (*WOX*) gene family transcription factors are broadly distributed in plant species and belong to the homeobox (HB) superfamily. Its members all possess a conserved DNA-binding homeodomain (HD) with 60 to 66 amino acid residues [54]. It has been reported that the number of *WOX* genes plays a significant role in a wide range of functions, including the maintenance of stem cells, embryonic development and polarization, and the development of lateral organs. In *Arabidopsis*, 15 *WOX* genes, *WUS*, and *WOX1-WOX14* have been identified and well-studied for their function. AtWOX1 plays a significant role in meristem development by regulating S-adenosylmethionine decarboxylase (SAMDC) activity or *CLV3* expression [55].

Moreover, the *WUSCHEL* gene (*WUS*) is required for the specificity and identity of stem-cells at the SAM [51,54]. *WUS* expression is crucial for meristem maintenance and shoot development [39], but the exact mechanism in which stem-cells are regulated is not fully understood. In the SAM, WUS, as bifunctional proteins, mainly act as a repressor but also becomes an activator when involved in the regulation of the *AGAMOUS* (AG) gene [56,57]. WUS directly represses the transcription of the *Arabidopsis Response Regulator (ARR-A)* genes, which encode for the intracellular inhibitor of cytokinin activity [58]. After the recognition of *ARR* as targets of WUS, a linkage between phytohormone (cytokinin) and the *CLV/WUS* stem cell network was established [59,60,61,62]. In *Arabidopsis*, Type-A ARRs are mostly transcriptional repressors, whereas type-B ARRs promote the cytokinin response and act as transcriptional activators [63]. Type-B ARRs, such as ARR1, ARR10, and ARR12, bind to the cis-element located in the promoter region of *WUS* and activate its expression. Type-A ARR negatively influences the meristem size, and WUS represses the expression of *ARR5*, *ARR6*, *ARR7*, and *ARR15* genes for proper meristem function [58,64]. *WUS* also protects apical stem cell meristems from differentiation by restricting the auxin signaling pathway via regulation of histone deacetylation [65]. To maintain the integrity of SAM, a high concentration of WUS protein repressed the expression of basic helix-loop-helix (bHLH) transcription factor HECATE1 (HEC1), which is expressed throughout the SAM, except for the OC [66,67]. HEC1 forms protein complexes with other bHLH transcription factors such as HEC2 and HEC3. HEC1 activates type-A *AAR7* and *ARR15* expressions and represses *CLV3* expression. Multiple feedback regulatory mechanisms mediated by transcription factors and hormonal components control *WUS* expression and meristematic fate in SAM [68,69,70].

It is also reported that *WUS* expression occurs in the OC, but in addition, it can also control the expression of *CLV3* in the CZ. The WUS protein migrates and binds to the *CLV3* promoter and thus regulates the expression of CLV3 [29,71]. Besides, computational modeling (Figure 1) shows that the WUS gradient is vital for the maintenance and regulation of the stem cell number [72,73]. Various reports have demonstrated that WUS moves to L2 and L1 via plasmodesmata under a highly regulated fashion, and this movement is required for WUS function and stem cell activity [74]. In *Arabidopsis*, WUS has been shown to interact with members of the HAIRY MERISTEM (HAM) family protein, GRAS-domain transcription factors [75], and *HAM1* and *HAM2* expressed in the RZ and lateral edges of PZ where *CLV3* expression is reduced. In contrast, *HAM1* and *HAM2* expressions are not detected in the CZ where *CLV3* is highly expressed. WUS activates *CLV3* only in the absence of HAMs; the apical-basal gradient of HAMs defines the pattern of *CLV3* expression domains [75,76]. The regulation of *CLV3* by *WUS* for the maintenance of stem cells is associated with the WUS gradient into the PZ, where the stem cell progeny is differentiated [18,77,78]. The CLV3-related signaling pathway in the stem cell domain is demonstrated by the diffusion of CLV3 peptide towards the inner layer of the meristem, where at least three receptor complexes recognize it. These receptor complexes include CLV1, CLV2, CORYNE (CRN), RECEPTOR-LIKE PROTEIN KINASE2 (RPK2), and BARELY ANY MERISTEM1 (BAM1/2) and represses the expression of *WUS* [42,43], which is crucial for the stem cell population [79].

The WUS activated gene is present in the central part of the SAM, and the repressed gene is located in the PZ. Additionally, in *Arabidopsis*, *WUS* represses those TFs-coding genes that are involved in differentiation, for example, *KANADI* (*KAN1*), (*KAN2*), *AS2*, *YABBY3* (*YAB3*), *KNAT1/BREVIPEDICELLUS* (*BP*), and *BELL1-LIKE HOMEODOMAIN5* (*BLH5*) [80,81]. The identification of these direct interactions can contribute to the understanding of the molecular network but with limitations to explain the mechanisms by which *WUS* controls stem cell homeostasis. If the stem cell progenitor is relocated beyond the stem cell niche, they then determine whether to be the part of the main axis or divide into lateral organs such as leaf primordia. The fate of distinction is primarily determined by the auxin influx carrier AUXIN RESISTANT1/like-AUX1 (AUX1/LAX) and the auxin efflux transporter PIN-FORMED (PIN1) (Figure 1). The PIN1 efflux carriers control the orientation of auxin transport into the neighboring cell and the concentration of auxin in the group of cells [82,83,84,85]. The auxin concentration may also vary even in the same group of cells. Several models have postulated phyllotactic patterning, which is mainly based on the interaction between auxin accumulation and distribution of the auxin efflux carrier (PIN1) [5,86]. Moreover, AUX1 is responsible for auxin accumulations, which are mainly in the L1 layers of cells. On the contrary, PIN1 drains auxin toward the base of the leaf primordium by inducing vascular tissue differentiation in the L2 and L3 layers [83,87].

### 1.2. Gene Functioning in Leaf Initiation

Leaf formation starts with the recruiting of founder cells in the peripheral of SAM by changing the pluripotent cell to a differentiated cell. The first step towards the emergence of leaf primordia is PIN1 mediated formation of auxin maxima in the PZ, and to repress the expression of *KNOX* genes [88]. This activity is necessary because the *KNOX* genes actively maintain the undifferentiated state of the cells [88]. A mutation in the *KNOX* genes changes the indeterminate cell to determinate. Auxin maxima enhance the growth and accelerate differentiation, while STM proteins have the opposite effect on both cell growth and differentiation [89]. The critical role of auxin in SAM is to control organogenesis and self-organization of SAM; auxin also specifies the organ primordium fate in the PZ of SAM [90,91]. Mutation in the auxin biosynthesis, transport, and signaling components affect plant growth and morphology. In *Arabidopsis*, 23 AUXIN RESPONSE FACTORs (ARFs) have been identified, and ARF works as a transcription factor that binds to the auxin-responsive elements (AuxREs) in the promoter region of the target gene [92]. Among other ARF transcription factors, ARF5 (which is also known as MONOPTEROS (MP)) is present in threshold form from PZ to CZ and plays an important role in gene expression to specify the meristematic and primordium fate [93,94]. The MP activates *ARABIDOPSIS HISTIDINE PHOSPHOTRANSFER PROTEIN6 (AHP6)* expression and inhibits *ARR7/ARR15* to control meristematic fate through the regulation of CK homeostasis [95]. Previously, it was suggested that MP was involved in stem cell regulation by inhibiting the expression of *DORNRÖSCHEN/ENHANCER OF SHOOT REGENERATION1 (DRN/ESR1)* that activated *CLV3* expression in the CZ [96,97].

The auxin-efflux protein PIN1 is expressed in the epidermal cell for auxin accumulation [81], which promotes the formation of pro-vascular tissue. PIN1 is the first marker for mid-vein formation before the leaf primordia bulge out [98,99]. A mutation in PIN1 can cause the irregular distribution of auxin and thus disturb the proper initiation of the leaf primordia [85,100]. In *Arabidopsis*, the *pin1* mutant blocked the floral primordium formation, and in tomato shoot apices treated with polar auxin transport inhibitor, such as N-1-naphthylphthalamic acid (NPA), abolished leaf formation [101,102]. The external application of auxin restored primordium formation in the NPA-inhibitor and *pin1* mutant, which indicates the importance of auxin in organ initiation [103]. In the peripheral of the meristems, organ primordia are separated and form a groove at the base that contains small slow-dividing and slow-expanding cells called the boundary domain [19,104]. The boundary domain is defined by several factors, including the NAM/CUC transcription factor family [25,105]. Mutants of *CUC* genes exhibited a lack of SAM and fused lateral organs, which indicated that this boundary domain contributes to meristem formation and enables the organ separation, as well as providing a particular status of hormones such as low brassinosteroid (BR) and auxin levels [105,106]. The *cuc3* loss of function mutant restrains leaf serration, which suggests the role of CUC3 in leaf serration. Moreover, the overexpression of *KNOX1* genes in lobed leaf phenotypes was suppressed in the *cuc2-3* mutant, which indicates that CUCs act downstream of KNOX-induced alteration in leaf morphology [46,107].

Cytokinin (CK) and gibberellic acid (GA) affect cell division, and *KNOX1* can increase the biogenesis of CK by up-regulating the *IPT7* genes that further block the *GA20-oxidase* gene required for GA biosynthesis (Figure 1) [108]. A high GA to CK ratio promotes determinacy; in contrast, a low GA to CK ratio facilitates the indeterminacy of the cells. Additionally, the KNOX-independent genetic pathway involves WUS and CLV3 that control the stem cells’ fates directly by regulating cytokinin-inducible response factors (discussed in the maintenance of SAM). A different pathway attains a low CK and a high GA, while determining how auxin promotes organ growth and is integrated with cell fate allocation by KNOX/AS1 protein is unclear. Furthermore, in the *as2* mutant background increases in the transcript level of *IPT3* and *KIP-RELATED PROTEIN2 (KRP2)* and *KRP5*, and this upregulation is a decline in introducing the double mutant of *ett, arf4* in the *as2* mutant background. Therefore, it is suggested that the expression of *IPT3*, *KRP2*, and *KRP5* is negatively regulated by AS1-AS2 through repression of *ETT/ARF3* and the *ARF4* function in the wild type [109]. In *Arabidopsis*, *KRP2* and *KRP5* encode for cyclin-dependent kinase inhibitors (CKIs), which is a key regulator of cell progression. During leaf formation, required cell proliferation is achieved by proper repressive control of *KRP2* and *KRP5* expression by AS1-AS2 [110].

In *Arabidopsis*, another pathway involves the MYB transcription factor encoding for *ARP* genes, which is expressed in the founder cells of the lateral organs, and represses the expression of the *KNOX1* gene that tends to promote leaf development (Figure 1) [31]. In *Arabidopsis*, the epigenetic repression of *KNOX* genes involves the binding of the AS1-AS2 complex to a specific sequence in the promoter region of the *BP* gene and inhibits its expression. It was confirmed that AS1-AS2 binds to specific sequences in the *BP* gene that works as a PRE (polycomb response element); the AS1-AS2 complex recruits the polycomb repressive complex 2 (PRC2), which is a stable silencer of STM regulators [111].

Furthermore, many reports indicate the precise regulatory mechanisms of *KNOX1* genes. In this regard, various genes (*AS1, AS2*, *SERRATE* (*SE*), *BLADE ON PETIOLE1* (*BOP1*) and *PICKEL* (*PKL*)) were found to be involved in the down-regulation of the *KNOX (KNAT1, KNAT2)* gene [32,112]. The mutants *as1* and *as2* have some similar phenotypic characteristics due to the ectopic expression of *KNOX1* genes and this expression results in the segmentation of the leaf primordium [32]. Moreover, SE was found to enhance the phenotypic expression of the *as1* and *as2* mutant. *KNOX* genes are normally regulated in both *pkl* and *se* mutant leaves; however, KNOX target gene *GA20ox1* is repressed, suggesting that PKL, SE and KNOX activities cover at least one specific target gene. SE represses a small family of micro-RNA targeted *Class III HD-ZIP* genes that promote meristem activity, the *se* mutant showed enhanced response to KNOX activity, indicating elevated level of *HD-ZIP III* expression, *HD-ZIP III* gain of function mutant reduced the expression of *GA20ox1*, mimicking effects of KNOX overexpression [113]. Class 1 *KNOX* gene overexpression prolongs the proliferation of the leaf cell within the lamina [114]. *BOP1* and *BOP2* are expressed at the proximal domain of the leaf primordium and activate the expression of the boundary specific *LATERAL ORGAN BOUNDARIES* (*LOB)* gene and adaxial expressed *AS2* gene [115,116]. The *bop1-1* mutant shows a similar effect with *as1-1* or *as2-2* and *stm-1* mutants, which suggests that BOP1 promotes or maintains a developmentally determinate state in leaf cells by regulating class 1 *KNOX* genes [101]. During leaf development, the expression domain of the *KNOX1* gene is also regulated by *ARP* genes and distinguishes the leaf founder cell from the meristem cell in SAM [33,117].

Besides, transcription factors, hormones, and mechanical forces are important for leaf development and morphogenesis. Turgor pressure and cell wall mechanics modulate the direction and rate of cell expansion and affect the pattern of plants. It has been confirmed that the cells in the CZ of SAM have stiffer cell walls compared to the PZ. Organ outgrowth is accompanied by an increase in cell wall elasticity [118]. It has also been shown that auxin regulates cell wall properties, and during leaf initiation, auxin not only reduces stiffness but also affects wall anisotropy through the modulation of the cortical microtubule dynamics [118].

## 2. Leaf Outgrowth and Expansion

Plants have the ability to grow indeterminately throughout their life and produce many repeated units during their lives. In addition, plants produce certain organs with determinate growth such as leaves, sepals, and petals. In this review, we focused on the size and growth of the *Arabidopsis thaliana* model plant in which the determined organ (leaf) is regulated by several genetic and environmental factors. The organs of the plants are formed from the reservoir of pluripotent cells, i.e., the meristem. Under the control of cell division and expansion, the leaf primordia achieves its natural size. As the leaf grows, the cell division at the distal end ceases and the expansion process occurs in which the cells at the base are strongly vacuolated rather than proliferated. The base displaces the older cells towards the distal end and eventually, they finally fall out of the proliferation zone. The cells (in a narrow region) are present between the blade and petiole junction, which bidirectionally divides the cells into two parts of the leaf. Furthermore, the dynamic of the cell proliferation regions rapidly appears at the “arrest front” boundary between the proliferation and expansion regions at a constant distance from the base of the leaf for several days before rapidly disappearing [119,120]. Thus, the two main processes of cell proliferation and cell expansion control the final leaf size, whereas any alterations in these processes may affect leaf formation. The timing of the transition from division to expansion within the growth of the leaf lamina is important to determine the final size, shape, flatness, and complexity. The class II TCPs (CINCINNATA-like TCPs) are the key regulators of timing from division to expansion [121,122]. The TCP family consists of plant-specific transcription factors; the CIN-TCPs is the subclass of the TCP family, which has a prominent role in controlling the transition from cell proliferation to expansion during leaf development. During leaf growth, the proliferation phase involves mitotic division interspersed with cytoplasmic growth to increase the primordium size due to the cell number. However, in the expansion phase, the organ increases the size by increasing the cell size [40,123]. There are many genes that positively regulate and control the transition from proliferation to expansion, for example, *AINTEGUMENTA (ANT)* [124], *KLUH/CYP78A5* [125] and *GROWTH REGULATING FACTORS (GRFs)* [126,127], whereas the negative regulators include CIN-TCPs and DAI [128]. In addition, the *PEAPOD1, 2 (PPD1, 2)* gene also acts as a negative regulator and changes the dispersed meristematic cells in leaf lamina to stomatal and vascular precursor cells [129]. The promotion of cell proliferation of class I TCPs depends on spatial and temporal expression domains, whereas the CIN-TCP represses cell proliferation (and so the loss of *CIN-TCP* gene functions increases the proliferation). In *Arabidopsis thaliana*, there are nine *GROWTH REGULATING FACTORS* coding genes (*GRF1-9*), and GRFs delay transition from proliferation to differentiation (Table 1).

TCP4 promotes the expression of *miRNA396*, which targets seven genes of the GRFs family (Figure 2) [127,130,131]. During the early stage of leaf development, *miR396* is expressed in the distal part of the leaf and the expression of *GRFs* is confined to the proximal regions and promotes cell proliferation. After cell proliferation, *miR396* is expressed throughout the organ to decrease *GRF* expression in the maturing organ [47]. How GRFs stimulate cell proliferation is not fully understood, but GRFs physically interact with GRF-INTERACTING FACTOR1/ANGUSTIFOLIA3 (GIF1/AN3), GIF2, and GIF3 to create a transcriptional module that regulates the leaf size by cell proliferation (Table 1) [41,127,132,133]. Contrary to normal plants, the *gif* mutant results in smaller and narrower leaves.

GIFs act as a transcriptional coactivator, which is associated with chromatin remodeling machinery. GIF1/AN3 promotes several proliferation-stimulating factors, such as ribosomes, to sustain the high demand of protein that is required in actively proliferating cells and represses the gene that promotes cellular differentiation [41,134]. AN3 lacks a DNA-binding domain, so GRFs help in the recruitment of the AN3-containing chromatin remodeling complex at the promoter region of target genes. During leaf development, the timing of declining GRF-AN3 abundance along the proximo-distal axes link with CIN-TCPs to create the mitotic cycle [40,134]. TCP4 accumulation starts at the tip of the primordium and then covers the whole actively dividing lamina such that TCP activity stops at the tip cells first and is slowly restricted to the base. The *miR319* is expressed proximal to the petiole so that miR319-sensitive CIN-TCPs form dynamic spatial gradients and the intensity tapers toward the tips due to inactivation in the distal cell and toward the base [135,136]. In *Arabidopsis*, the *jagged and wavy Dominant (jaw-D)* mutant exhibits highly crinkly shaped leaves because of the overexpression of *miR319A*. The miR319A down-regulates *CIN*-like *TCP* genes and their transcript level is dramatically reduced in jaw-D because of the ectopic expression of *miR319A* (Figure 2) [137,138]. The coordination between the miRNA319-TCP and miR396-GRF modules controls the marginal and overall growth of the leaf through the regulation of cell proliferation [139,140]. The second pathway that controls the leaf lamina outgrowth is the *ANT* and *AINTEGUMENTA LIKE (AIL)* gene, which encodes for an AP2/ERF transcription factor. CIN-TCP and ANT have antagonistic effects on the G1/S transition of the cell cycle, and the transcription of *CYCLIN-DEPENDENT KINASE INHIBITOR 1 (ICK1)* is activated by CIN-TCP that interacts and blocks the activity of G1 cyclins whose transcription is enhanced by ANT [111]. ANT likely functions downstream on auxin, and the auxin-inducible gene *ARGOS* (an auxin-regulated gene involved in organ size) encodes the ER-localized protein, which is a protein of unknown function. The overexpression or suppression of *ARGOS* alters the aerial part, such as leaves, flowers, and siliques. The difference in size is mainly due to alterations in cell numbers and the duration of cell proliferation periods. In *Arabidopsis*, *ARGOS-like (ARL*) genes, which have some sequence homology to the *ARGOS* gene, are responsible for the promotion of organ growth via cell expansion. Similarly, the kinase encoded by the *TARGET of RAPAMYCIN (TOR)* gene is required for cell expansion in the leaves and its overexpression causes an increase in leaf size due to cell expansion [141,142].

ARGOS enhances the expression of *AINTEGUMENTA* (*ANT*), another size of the regulator gene, and the change in *ANT* function has similar effects as *ARGOS* expression. ANT maintains the expression of D-type cyclin CYCD3;1. The losses in D-type cyclin may cause premature termination of the proliferation phase and change the overall size of the leaf [143]. The ANT also interacts with another family of transcriptional regulators, i.e., ARFs. The ARFs mediate the auxin response and limit the cell size (proliferation) by repressing the *ANT* and CYCD3;1 activity [144,145].

In addition, when the functions of four NGATHA (NGA) transcription factors are lost, enhanced leaf marginal growth and serration are observed. However, the overexpression of NGAs reduces marginal growth and indicates the redundant roles of NGAs in controlling the switch between leaf marginal expansion and differentiation [140,146]. In *Arabidopsis*, the transcription factors of NGAs and TCPs terminate the blastozone meristems by inhibiting the expression of the *WOX* gene [140,147,148]. According to these observations, the leaf blastozone appears by *PRESSED FLOWER (PRS)* expression and is restricted to the overall marginal regions of young leaves and further restricted to the proximal regions of older leaves [140]. Besides, the genes that encode homeobox transcription factors, *WOX* genes, *PRS/WOX3*, and *WOX1*, redundantly promote leaf blade outgrowth [140,149]. WOX1 and PRS also promote the expression of *KLUH*, which encodes a cytochrome P450 CYP78A5 monooxygenase that promotes cell proliferation in a non-autonomous manner (Figure 2) [48]. The loss of function mutant *kluh* produces smaller organs due to the premature arrest of cell proliferation, while the overexpression of *KLUH* produces larger organs with more cells; and therefore, KLUH promotes organ growth. KLUH is involved in generating mobile factors, and the computational analysis showed that the tissue polarity system specifies the growth patterns in developing leaves [150]. In the early developmental stage, the basic pattern of the growth rate was established across the leaf. Previous studies indicated that leaf meristem activities are controlled by both local regulations and by mobile growth factors that function at the organ level [151].

Besides, in the regulation provided by miRNA-transcription factors, the hormonal control also involved regulating the leaf size and intercalary growth. The most prominent hormones regulating the leaf size are gibberellins (GAs) and brassinosteroids (BRs), which promote leaf growth by cell proliferation and expansion. The GAs increase cell proliferation by repressing cell cycle inhibitors such as KIP-RELATED PROTIEN2 (KRP2) and SIAMESE [152,153]. Overexpression of the BR biosynthesis gene *DWARF4* or BR receptor-encoding gene *BRASSINOSTEROID INSENSITIVE1* results in larger leaves [128,154]. The BR also represses *PPD1* and *PPD2*, which encodes the transcription factor that limits meristemoid cell proliferation [155]. BRs promote cell expansion via an antagonistic trio of bHLH transcription factors, which are also regulated by GAs, light, and temperature (Figure 2) [156,157].

In the distal region of the leaves, the epidermal and mesophyll cells still have some precursor cells to generate the stomata as well as vascular tissue after cell proliferation ends [158]. These cells can divide the distal region, which is known as dispersed meristematic cells (DMSCs). The DMSCs proliferation is under the control of putative transcription factorPPD1, PPD2. A double mutant of *ppd1* and *ppd2* produces bell-shaped leaves due to the proliferation of the cell, especially the marginal cells [155].

## 3. Adaxial/Abaxial Patterning

SAM provides a site for lateral organ formation into the primordium and develops into a flat structure. The lateral organs (leaf) establish polarities along the adaxial/abaxial, mediolateral, and proximodistal axes. The development of adaxial/abaxial patterns requires precise coordination between hundreds of cells throughout primordium development. After the establishment of adaxial/abaxial patterns, they provide a cue for further asymmetric growth. The acquisition and maintenance of adaxial/abaxial polarity are driven by regulatory networks of genes and highly conserved transcription factors [159].

The two sets of transcription factors are expressed at the sides of the leaf, i.e., the upper and bottom that work antagonistically to control the adaxial/abaxial polarities of the leaf. In *Arabidopsis thaliana*, the members of the HD-ZIP III family (including *PHABULOSA (PHB)*, *PHAVOLUTA (PHV)*, and *REVOLUTA (REV)*) specify the adaxial cell fate of the leaf (Table 1) [160]. Many transcription factors, i.e., Myb transcription factors PHANTASTICA (PHAN)/AS1 form complexes with the transcription factor AS2 of the LOB to specify the adaxial fate [36,161,162]. The abaxial specification includes three members of the *KANADI* gene family (*KAN1*, *KAN2*, and *KAN3*) [39,163], four members of the *YABBY (YAB)* gene family (*FILAMENTOUS FLOWER (FIL)* [52], *YAB3*, *YAB5*, and *YAB2*)), and two *AUXIN RESPONSE FACTORS* (*ETTIN ETT/ARF3* and *ARF4*) [53,164]. These regulators, except *AS1/PHAN*, are specifically expressed in the abaxial or adaxial side and their mutual regulation is very important for proper establishment and maintenance of adaxial-abaxial patterns. In *Arabidopsis thaliana*, the dominant mutants, *phv* and *phb*, form rod-shaped leaves that result in adaxialization of the leaf in the circumference. In fact, the adaxial fate controlling genes, *HD-ZIP III*, are repressed by miRNA165/166 [165]. Ectopic/constitutive overexpression of *miRNA165* and *miRNA166* can reduce the transcript level of *HD-ZIP III* genes [166]. The *KANADI* gene family requires the abaxial identity of leaves that encode nuclear-localized GARP domain transcription factors [167]. The function of KAN1 disturbs the adaxial/abaxial polarity of the leaf. Ectopic/constitutive expressions of the *kan1* mutant with the 35 s promoter produces a narrow cotyledon but no subsequent leaf production [168,169,170]. In fact, *PHB* is expressed when *KAN* genes antagonistically regulate *HD-ZIP III* genes [171]. *KAN1* directly represses the expression of *AS2*, and *AS2* indirectly represses the expression of *KAN1*. Furthermore, *AS2* also negatively regulates *ETT*, *KAN2*, and *YAB5*. Likewise, the opposing effect of KAN and HD-ZIP III on the auxin biosynthesis gene has been confirmed by many researchers. A recent study corroborated the finding that the *KAN1* allele is responsible for suppressing the expression of PIN1. The ectopic expression of *KAN1* reduces the gene expression of pro-cambium cells and PIN1 in pro-vascular cells. *APUM23* is a new regulator of leaf polarity, which encodes approximately 20 PUF RNA-binding proteins in *Arabidopsis thaliana* and interacts with leaf polarity that is required for the maintenance of genes [172,173]. Another gene family, which specifies the abaxial cell fate, is the ARFs, which binds to the promoter element of the auxin response gene and transducer auxin signaling, and their role in adaxial/abaxial polarity is confirmed through mutation and up-regulation patterns of *ETT (ARF3)* and *ARF4* [174,175]. The polarity defects were visualized in *ett-1*, *arf4-1*, and *ett-1*, and *art4-2* double mutants. In these mutants, the produced abaxialized leaves were similar to *kan1* and *kan2* mutants. The direct interaction between *KAN* and *ARF* indicates that overlapping patterns control the polarity of the leaf. Both the *ETT* and *ARF4*, as a target of TAS3, are derived from trans-acting siRNAs (ta-siRNAs) that up-regulate small interfering RNA (siRNA) [98,176]. According to this experiment, ta-siRNA insensitive *ETT* or *ETT* overexpression in the *rdr6-15* mutant background showed a defect in leaf morphology. These observations specify TAS3 ta-siRNAs as a negative regulator of abaxial cell fate through targeting of *ETT* and *ARF4* gene expression in small RNAs [98,177].

The adaxial and abaxial pattern is important for lamina outgrowth and started from the boundary of adaxial/abaxial to the medial/lateral axis. The leaf primordia represent another meristematic zone called the plate meristem or blastzone. Many genes are involved in the medial/lateral specification, but the most important is the *YABBY* gene family, which encodes for proteins with a zinc finger and a helix-loop-helix domain, and also plays a key role in lamina outgrowth. The *Arabidopsis* genome contains six *YABBY* genes (*FIL, YABBY2 (YAB2), YAB3*, *YAB5, CRAB CLAW (CRC)* and *INNER NO OUTER (INO)*) [178,179], and four of these are expressed in the vegetative primordium and (*FIL*, *YAB2*, *YAB3*, and *YAB5*) and the other two are expressed in floral organs. Consistently, the YABBY expression is localized between the adaxial and abaxial side at the leaf margin and promotes lamina outgrowth. Interestingly, the *YABBY* gene expression is regulated by the members of the polarity pathway such as the HD-ZIP III, AS, and KANADI pathways [180,181,182,183]. In the *kan1* and *kan2* double mutants, the *YABBY* gene is required for ectopic outgrowth. Therefore, for lamina outgrowth, *YABBY* gene function is integrated with polarity signals [184]. The *YABBY* gene is important for repression of the genes of SAM in developing leaves and promoting maturation of the leaves. In the double mutants of *fil* and *yab3*, class 1 *KNOX* genes are ectopically expressed [185,186,187]. Four *YABBY* gene mutants expressed in the leaf primordium show narrow leaves but only limited defects on leaf polarity.

For the *WOX* genes, at least two subfamily members are essential for lamina outgrowths such as *WOX1* and *PRS/WOX3*. In *Arabidopsis*, *PRS* is expressed in the margins of the developing leaf primordium; the *prs* mutant causes the deletion of stipules at the base of leaf margins without a reduction in leaf width and due to the genetic redundancy with *WOX1* [188,189]. The *WOX1* gene is expressed along with the adaxial-abaxial juxtaposition and overlaps with *PRS* at the marginal region of the leaf. The double mutant of *wox1/prs* showed prominent defects in the lamina outgrowth and redundantly acted to enhance the leaf width. It has been identified that *WOX1* expression occurs around the meristem at the boundary between the HD-ZIP III and *KAN* expression domains. In the *kan1* and *kan2* double mutants, the expression of *WOX1/PRS* is enhanced in the abaxial domains of the leaf, which suggest that the *KAN* gene may function to negatively regulate the expression of *WOX1/PRS* [190]. Therefore, misexpression of *WOX1/PRS* may explain the ectopic formation of abaxial margin-like outgrowth that occurs in the *kan* mutants. The expression of *WOX1/PRS* is negatively regulated by AS2 in the adaxial domain of the leaf. *AS2* expression is also repressed by WOX1/PRS to restrict its expression on the adaxial side of the leaf (Figure 2). Meanwhile, abaxial-specific gene expression is also influenced by WOX1/PRS and these interactions help to restrict *WOX1/PRS* expression toward the margin domain of the leaf [190,191].

In the marginal regions of the leaf, *WOX1/PRS* is expressed and enables flattening of the leaf. MP and auxin act together as a positional cue for patterning the *WOX1/PRS* marginal regions [179]. Abaxial factors, such as KAN, restrict the marginal domain expansion [48], which binds to the same elements as an MP in the *WOX1* gene promoter and inhibits their expression [167]. During leaf development, the auxin maxima are first formed at the tips of young primordia and promote distal growth. Previous evidence suggests that auxin works downstream of the leaf polarity genes and enhances lamina outgrowth. In the triple mutant of *kan1, kan2* and *kan3*, there was enhanced lamina outgrowth on the hypocotyl due to ectopic localization of PIN1 proteins [142]. The distribution of auxin and PIN1 is greatly influenced by *yabby* mutants. The *as1* and *as2* mutant showed asymmetrical lamina outgrowth due to the asymmetric distribution of auxin. Recent studies revealed that the *YUCCA (YUC)* gene, which encodes catalyzation, is involved in auxin biosynthesis and plays an important role in lamina outgrowth and leaf margin formation. In *Arabidopsis*, there are 11 *YUC* genes that have been identified; the mutant of at least four genes led to loss of marginal characters. Interestingly, *YUC* gene expression is enhanced in ectopic lamina outgrowth, which is due to ectopic adaxial and abaxial juxtaposition in the *as2 rev* and *kan1 kan2* double mutants [192,193].

## 4. Leaf Margins Serrations

The leaf is developed from the peripheral zone of the SAM under the control of many intrinsic and extrinsic factors. The leaves are different in size and form, which is a complicated process. The various forms of leaf margins include entire, serrate, and lobe margins. The molecular mechanism of the *Arabidopsis* leaf indicates that the leaf serrations get more pronounced during plant development. Additionally, miRNA164A, CUC2, PIN1, and DPA4 regulate the leaf serration [45,46,194]. In *Arabidopsis thaliana*, leaf serration is less prominent in rosette leaves compared to leaves that develop later. Leaf serration is also different in different accessions of *Arabidopsis thaliana*. The optimal expression of the genes is necessary for continuous growth and development. The miRNAs mainly contribute to the regulation of gene expression [45,195]. In leaf serration, the boundary controlling gene *CUC1*, *CUC2*, and *CUC3* is required for the maintenance of the shoot apical meristem [45,195]. Furthermore, reports indicate that CUC2 has a very important role in leaf serration. The members of the NAC transcription factors can be expressed in the boundary region and suppress growth. A mutation in this gene at an early stage produces a cup-shaped fused cotyledon. The *CUC1* and *CUC2* gene expression are controlled by miRNA164A [45,46,196]. The ectopic expression of *miRNA164A* represses *CUC2* gene expression; therefore, the leaf is less serrated (unlike a wild type). The mutations in the *mir164a* produce deep serration and contrast with the overexpression of *miR164A*, which causes smooth leaf margins [45]. In overexpression, the *CUC2* expression is high and plant leaves are highly serrated (Figure 3). CUC2 promotes PIN1 efflux auxin carriers [4]. The mutant of the *pin1* genotypes causes inhibition of the auxin efflux carrier and the leaf has smooth margins [65,197]. The other plant hormones, such as cytokinin (CK), gibberellins (GA), and many others, also have an important role in organogenesis depending on its concentration [198]. *JAGGED LATERAL ORGANS* (*JLO)* is a member of *LATERAL ORGAN BOUNDARY DOMAIN* gene family and is transiently expressed at the site of organ initiation [199,200], promotes the *PIN* expression for auxin maxima, and resolves in the leaf-meristem boundary during outgrowth [103,200,201].

In addition, the *cin-tcp* mutant displays prolonged marginal growth leading to the formation of the lobe and crinkly leaves due to ectopic expression of boundary specific *CUC2* genes and meristem-specific *KNOX* genes [196,202]. CIN-TCPs repress *CUC* activity through miR164 that targets CUC2. Additionally, TCP4 interacts with CUC2 and CUC3 to prevent their dimerization and transactivation potential in the juvenile stage to inhibit leaf serration [196]. However, in the later stage, the SPL transcription factor destabilized the interaction of TCP-CUC from relieving CUC protein from inhibition. Therefore, the *spl* mutant loss of function and *CIN-TCP* gain of function both have the same reduced organ size, which suggests that SPL interferes with CIN-TCP in growth repression [203,204].

The *DEVELOPMENT-RELATED PcG TARGET IN THE APEX4 (DPA4)* negatively regulates the expression of *CUC2* independent of MIR164A and modulates leaf serration [151]. The TCP interactor containing EAR motif protein1 (TIE1) is a transcriptional regulator located in the nucleus. TIE1 recruits co-repressor TOPLESS (TPL)/TOPLESS-RELATED (TPR) in leaf margin morphology development and inhibits the activity of TCP at the protein level. Therefore, over-expression of *TIE1* can cause an increase in leaf serration [151,205]. TIE1 and TEAR1 (TIE1-ASSOCIATED RING-TYPE E3 LIGASE1) have a mutual role; a mutation in *TEAR1* and its homologous genes increases leaf serration [206]. TIE1 recruits TPL/TPRs to inhibit TCP activity, and TEAR1 restricts the inhibition of TCP through the degradation of TIE1. *TIE1* and *TEAR1* indirectly affect the leaf margin by regulating TCP (Figure 3).

A peptide in the plant called EPFL2 (Epidermal Patterning Factor-Like) and members of the *ERECTA (ER*) family are also involved in the morphogenesis of leaf marginal serrations [207]. When the *EPFL2 or ER* is mutated in *Arabidopsis*, the leaves become smooth and the auxin is detected throughout the leaf margins. EPFL2 forms a ligand-receptor pair with ERECTA and thereby inhibits the response of auxin at the leaf tooth area, which in turn inhibits the expression of *EPFL2* and forms a negative feedback loop [207]. This feedback system maintains the auxin response pattern during leaf margin growth. In addition, current studies have shown that *JAGGED (JAG), JLO*, and Trifoliate can also regulate leaf margin morphogenesis through the auxin pathway and also affect the *KNOX* regulatory pathway (Figure 3) [199].

BLADE ON PETIOLE1 (BOP1) and BOP2 belong to BTB family proteins, which form dimmers that function as transcriptional activators. *BOP1* and *BOP2* are expressed at the base of lateral organs, and *BOP* expression at the base of the leaves can directly regulate *AS2* and inhibit the expression of the *KNOX* gene [116]. The leaves of *bop1* and *bop2* double protrusions become larger, and leaf teeth and leaf fins grow at the petioles [208,209]. Therefore, BOP ensures normal leaf morphogenesis by inhibiting *KNOX* expression at the leaf base and petiole [153,210,211]. High auxin accumulation causes more cell division and growth, and consequently, the teeth regions of the leaf have high auxin maxima [102,212,213]. Auxin represses the expression of the *CUC2* gene, and therefore, *CUC2* expression is restricted to the sinus region of the leaf [45,46]. CUC2 also suppresses the growth of cells in the sinus region and thereby promotes tip outgrowth. The exact mechanism of the CUC2 and auxin interaction is still not fully understood.

## 5. Developmental Functions of Micrornas

MicroRNAs are a group of non-coding RNAs that play an important role in diverse cellular pathways and regulate most of the plant and animal biological processes [214]. The transcription and maturation of microRNAs involve a series of complex processes. In the first phase, endogenous genes are transcribed by Pol II or III into long primary miRNAs that consist of several hundred nucleotides. Afterward, single strand pri-miRNAs are folded to form a hairpin-like secondary structure [215]. Pri-miRNAs are processed by endonuclease RNAase III and the mechanism is different in plants and animals [216,217]. Plants lack Drosha homologs and after the pri-miRNAs formation, the RNase III enzyme DICER-LIKE1 (DCL1) regulates the first and second steps (Figure 4). In contrast, in animals, the first step involves Drosha, which cuts miRNAs strands, and in the second step, the pre-miRNAs are processed through Dicer with the aid of HYL1 and SE to form a duplex in the nucleus. The mature microRNAs duplex consists of active and complementary strands. The active strands are called guide strands while the complementary strands are called passenger strands. The guide strands with lower thermodynamic stability and high abundance are loaded into ARGONAUTE (AGO)-associated RNA-induced silencing complexes (RISCs) and target the mRNA transcript [218,219]. The passenger strands of miRNAs become degraded, and the accumulation of passenger strands is lowered by guide strands [220,221]. Many studies have confirmed that miRNAs act as regulatory factors in a large number of biological processes and consist of different numbers of miRNAs in a single species [222,223]. The miRNA biogenesis mutants (*dcl1*, *hyll*, *se*, and *hen1*) and *ago1* mutants produce developmental defects [224]. In this review, we described some of the miRNAs and their function in different stages of plant development, including phase transitions, hormone biosynthesis and signaling, pattern formation, and morphogenesis. During leaf development, different types of miRNAs have a prominent and important role that modulates leaf development in different phases such as establishments, transitions, modifications, and senescence. Transcriptome profiling data showed that several miRNAs are involved in early embryonic development such as miR156, miR166, miR167, miR390, and miR394 [225]. The interaction between two miRNAs affects the maintenance of the meristem and leaf initiation, such as the interplay of LEAF CURLING RESPONSIVENESS (LCR) and the miR394 mediated non-cell-autonomous network, and the module of miR160 and miR165/66 mediated cell-autonomous pathways [165,226]. The protoderm-specific miR394 confers stem cell maintenance by repressing the gene expression of *LEAF CURLING RESPONSIVENESS (LCR)*, which also regulates a local feedback loop mediated by *WUS* and *CLV* genes [226].

Small RNAs, such as miR169, miR167, and tasiR-ARFs, target the *ARF* genes, which play an important role in auxin signaling. During shoot and root development, miR160 regulates the expression of *ARF10*, *ARF16*, and *ARF17* [227]. However, overexpression of *miR160* or *miR160-*resistance ARFs leads to pleiotropic and developmental defects in all aerial organs [228,229,230]. Besides, miR167 targets *ARF6* and *ARF8*, which redundantly regulates ovule and anther development [231]. During the establishment of leaf boundaries, both miR165/166 and miR390 move between cells and interact with each other. In both *Arabidopsis* and maize, miR165/166 targets homeodomain-leucine zipper transcription factor genes to establish the abaxial identity of lateral organs. The abundantly produced miR165/166 in the abaxial side of the leaf targets the mRNAs in the adaxial side where these genes specify the adaxial characteristics (Figure 2) [232]. In *Arabidopsis*, miRNA resistant varieties of *PHB* and *REV* genes lead to adaxializtion of the leaves. The tasi-RNAs from the *TAS3* locus target *ARF3* and *ARF4*, which results in the abaxial differentiation of lateral organs, vegetative traits, and leaves [39,233,234]. The tasiR-ARFs repress the expression of *ARF3* and *ARF4*, which contributes to adaxial specification in different species (Table 2). However, mutations in ta-siRNA biosynthesis do not exhibit any prominent change in leaf polarity, which is most likely due to the existence of a parallel mechanism to control adaxial polarity [182,235]. To develop a leaf bladeless (*lbl*) mutant in maize, a homolog of *Arabidopsis* SGS3 requires ta-siRNA biogenesis that promotes the abaxialization of leaves. The two small RNAs, miR165/166, and tasiR-ARF are expressed in the opposite side of the leaf and establish the abaxial-adaxial axis in leaf development [236,237]. In addition, during further modification, divergent leaf growth polarity is strongly correlated with the miR396-*GRF* expression gradient. The miR164 regulates the expression of *CUC1* and *CUC2* transcription factors, which are very important for the proper establishment of organ boundaries, floral patterning, and leaf morphogenesis throughout plant development [45,238].

The most conserved miR156 found in all land plants in *Arabidopsis*, miR156/157 targets 10 of the SPL TFs, which promote vegetative phase changes and floral transition [263]. *SPL3/4/5* determines trichome formation and distribution, *SPL9/15* modulates adult leaf morphology, and *SPL2/10/11* regulates the lamina shape and acts independently of miR172 (Figure 4) [264]. Overexpression of miR156 prolongs the juvenile stage and extremely delays the flowering stages. In *Arabidopsis*, miR159 targets *MYB33*, *MYB65*, and *MYB101* genes that activate gibberellin-responsive genes in the aleurone layer during germination. The overexpression of miR159 is responsible for the delays in flowering [165]. In *Arabidopsis*, miR172 regulates six of the AP2-domain transcription factors, including [240], TARGET OF EAT1, TOE2, TOE3, SCHLAFMÜTZE (SMZ), SCHNARCHZAPFEN (SNZ), and promotes flowering by repressing *TOE1* and *TOE2* [264,265]. The miR156 and miR172 act in a linear pathway and coordinate vegetative and floral transition (Table 2) [264]. In a petunia hybrid and *Antirrhinum majus*, miR169 controls the spatial restriction of the homeotic class C genes that are required for the identities of reproductive organs in the flower [251,266]. In an ectopic expression, miR169 transforms petals into stamens and targets the *NF-YA* genes that are activators of class C gene expression. In *Arabidopsis*, miR169 restricts the class C gene expression to delegate the transcription factor gene of *APETALA2* (AP2) [253]. In *Arabidopsis*, miR172 targets the *AP2* mRNA, which is uniformly present in all four floral whorls, unlike other floral homeotic genes that are confined to two whorls [253,254]. However, miR172 mediates regulation of AP2 at the translational level and that is why AP2 protein is more concentrated in the outer two floral whorls [215,267]. In *Arabidopsis*, miR319 targets five *TCP* genes that mostly control cell division during leaf development [125,268].

Senescence is the last stage of leaf morphogenesis that involves many distinctive actions [103,269]. In *Arabidopsis*, miR164 and miR319 modulate aging-induced cell death and leaf senescence. The miR164 target *ORESAR1 (ORE1*) functions as a positive regulator of senescence. The central components of ethylene signaling *ETHYLENE INSENSITIVE2* and *3 (EIN2/3)* induce *ORE1* in an age-dependent manner [270]. During the early stage of leaf development, miR164 down-regulates *ORE1* expression but also up-regulates at a later stage by EIN2/3. In addition to ethylene, other plant hormones, such as auxin and JA, also play important roles in controlling plant senescence. ARF2, a negative regulator of auxin responses, is believed to regulate leaf longevity [269,271]. The miR393 targets TIR1, which are the auxin receptor and F-box genes that play an important role in auxin biosynthesis [259]. Furthermore, miR319 targets the *TCP4* genes [136,272], and TCP4 has been shown to activate jasmonic acid (JA) biosynthesis gene *LIPOXYGENASE2*, which increases the level of JA and accelerates leaf senescence [273,274].

## 6. Conclusions and Future Perspectives

The developmental transition during shoot development in plants is regulated by factors that originate outside and within the shoot apical meristem (SAM). Lateral organ initiation at the shoot apical meristem involves a complex mechanism of hormones and downstream transcriptional regulation that leads to the formation of different organs such as the leaf, shoot, and flower. From vegetative to reproductive transition, the leaf-derived external signal and internal factors that cause the vegetative phase change and that are spatially coordinated remain unclear. This review elucidates the interaction of different factors (genes, miRNAs, and hormones) that maintain the STM cells’ identity. For example, the homeobox gene *WUS* is expressed at the organizing center (OC) and regulates the boundaries of the STM cell niches. These factors not only maintain the stem cell identity but also help in the initiation, growth, and adaxial/abaxial patterning of leaf development. It has also been suggested that many small RNAs help in early leaf development because of their ability to clear out the transcript when the cell passes from one stage to another. Subtle and complex mechanisms such as leaf development require many levels of control, for instance, buffering and plasticity of the small RNAs is one example of how plants achieve this control. Elaborate signaling and effector networks are also involved in leaf development and further study of these aspects is warranted.

## Figures and Tables

**Figure 1 ijms-21-05132-f001:**
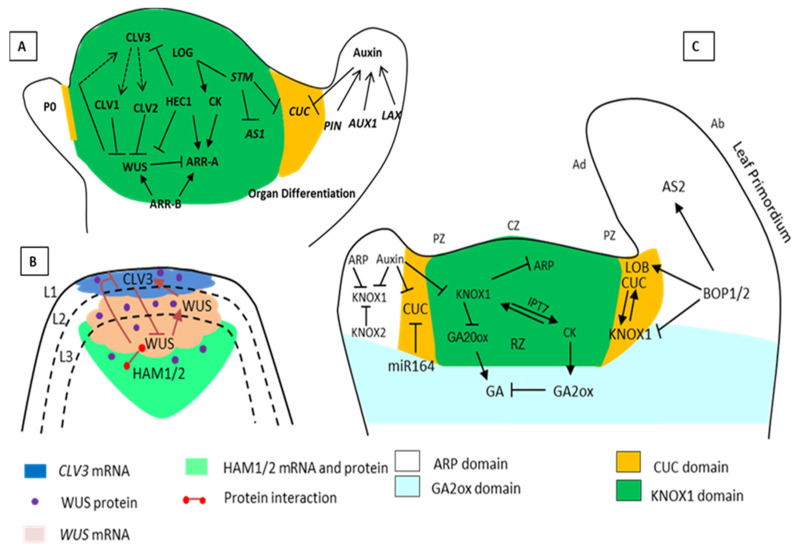
Schematic representation of Shoot Apical Meristems (SAM) maintenance by the various interacting genes. (**A**) The pluripotent stage and a specific number of cells in the SAM are controlled by the (WUS/CLV3) negative-feedback loop. CLAVATA3 (CLV3) is a ligand attached to CLV1 and CLV2 for the restriction of WUS. However, WUS activates CLV3 and works as a stem cell-promoting protein. STM activated the biosynthesis of CK through LONELY GUY (LOG). The STM and KNOX related genes also keep the stem cells undifferentiated by suppressing the expression of AS1 and gibberellic acid (GA) biosynthesis. STM also restricts the expression of the CUC gene due to negative regulators in a specific area. (**B**) The HAM and WUS-CLV3 loop. The regulatory loop requires CLV3, WUS, and HAM; the CLV3 negatively regulates the WUS expression, and the WUS protein moves from the organizing center to the active central zone (stem cells) to activate CLV3 expression. HAM1/2 is an interacting partner of WUS, and together with WUS protein suppresses the expression of CLV3 in the rib meristems. The expression zone of the CLV3 gene (blue), WUS gene (pink), HAM proteins (green) and the dot marks the WUS protein. (**C**) Regulatory networks that control leaf initiation. The cells in the SAM are arranged into layers L1, L2, and L3 and further into a distinct group of either tunica or corpus. According to the expression of genes, the SAM architecture is organized with the central zone (CZ), peripheral zone (PZ), organizing zone (OZ), and rib zone (RZ). During leaf initiation, auxin maxima repress the expression of the KNOX1 domains (gene) indeterminate meristem domains. KNOX1 maintains a high level of CK and low levels of GA in the meristem. In the ARP domain that has leaf identity, the leaf primordium separates from SAM by expression of boundary specific genes CUC and BOP regulates the petiole specification and polarity (positive and negative regulations are indicated by pointed and T-shaped arrows).

**Figure 2 ijms-21-05132-f002:**
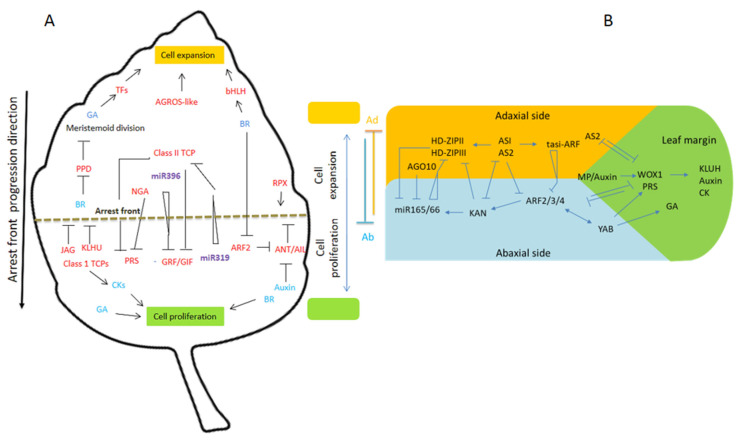
Regulation of *Arabidopsis* leaf proliferation and cell expansion transition. (**A**) the cell cycle arrests the front migration basipetal growth gradient from the apex to the base of the blade petioles, which is a process that is promoted by miR319-TCP modules and repressed by miR393-GRF modules. The gradients of miR319 and miR396 are complementary to TCP and GRF expression. TCP and NGA repressed the blastozone activity. During intercalary growth, the GAs and BRs promote both cell proliferation and expansion (positive and negative regulations are indicated by pointed and T-shaped arrows). Red, genes; blue, hormones; purple, small RNAs. (**B**) Shortly after its initiation, the young leaf primordium has three regions: adaxial, abaxial, and leaf marginal regions. These regions are determined by the region’s specific transcription factors, such as HD-ZIP III, KANADI, and PRS WOX1, involving multiple negative feedback loop mechanisms. For the sake of clarity, all the interactions are not shown here. Adaxial-abaxial polarity is determined by highly interconnected gene networks. In this network, tasi-ARF limits abaxial determinants and AUXIN RESPONSE FACTORS ARF3/ARF4 to the bottom side. In addition, miR165/166 restricts the expression of adaxial determinants HD-ZIPIII to the top of the leaf (the positive and negative regulations are indicated by pointed and T-shaped arrows, respectively).

**Figure 3 ijms-21-05132-f003:**
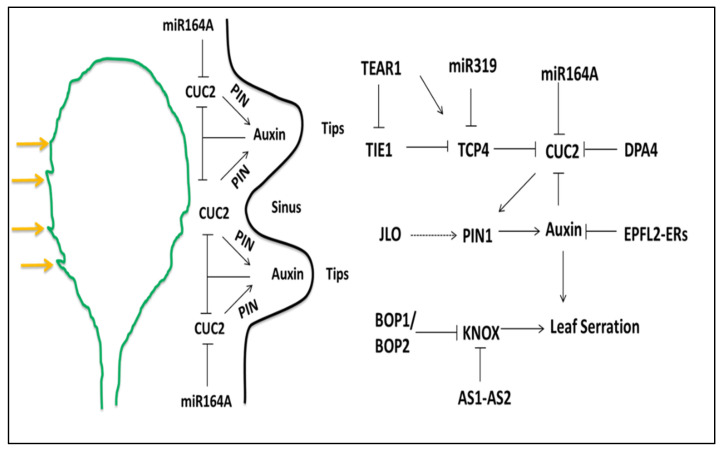
Schematic depiction of the factors involved in the modification of leaf margins, the genetic interaction between different genes and hormones cause the formation of leaf serration. The CUC2 gene promotes the auxin maxima via PIN1 efflux protein in the epidermal cell. The high concentrations of auxin at the tips block, the expression of CUC2 and thus, the expression of CUC2 gene is restricted to the sinus region of the leaf; miR164A also suppresses the CUC2 expression. The other regulatory factors shown in the figure is discussed in detail in the text (the positive and negative regulations are indicated by pointed and T-shaped arrows, respectively).

**Figure 4 ijms-21-05132-f004:**
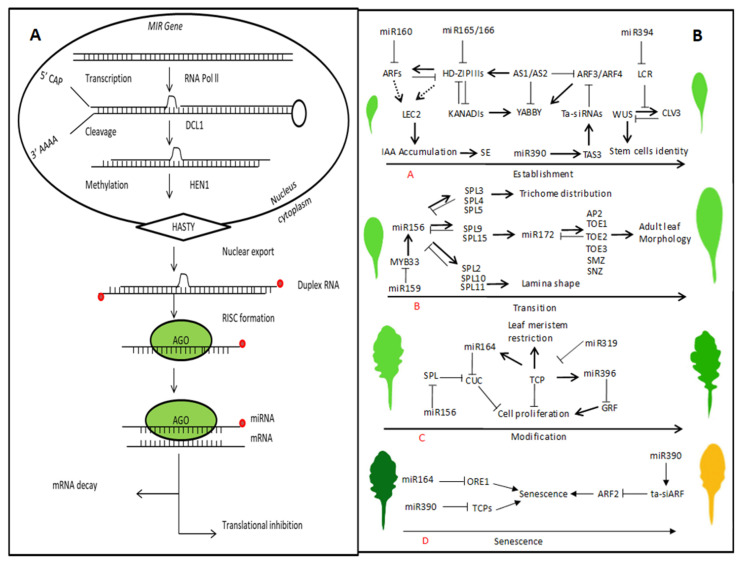
A schematic pathway for miRNA biosynthesis and the degradation of target mRNA. (**A**) Primary miRNAs are formed by polymerase II that folds back to form a hairpin structure. Splicing and further processing in the nucleus include the interactive role of different proteins and the CAP-binding proteins CBP20 and CBP80. Further processing occurs by DCL1 and forms miRNA-miRNA* duplexes, which are methylated by HEN1 and transported to the cytoplasm. The mature miRNA is unwound to yield a 22-nt single-strand and incorporated into AGO1-containing RISC (RNA-induced silencing complex) and other regulatory proteins that directly inhibit the translation or cleavage of the target mRNA transcript. (**B**) Leaf development is a complex biological process with multiple regulatory networks regulated by miRNA and their target genes. The leaf development consists of different stages such as establishment, transition, modification, and senescence. The miRNAs play important roles in each stage. The miR160 and miR165/166 control leaf initiation; leaf polarity is determining by miR165/166 and miR390; leaf morphology is regulating by miR164 and miR319; phase initiation is determined by miR156 and miR172; leaf senescence is determined by miR164 and miR319 (the arrows indicate positive and T-shaped indicate negative regulation).

**Table 1 ijms-21-05132-t001:** A list of the key genes involved in leaf growth and differentiation.

Gene	Description	Biological Function	Species	References
*ASYMMETRIC LEAVES1/ROUGH SHEATH 2/PHANTASTICA (ARP)*	MYB domain protein	Stem cell differentiation	*Arabidopsis*, *Zea mays*, *Antirrhinum majus*	[31,32,36,37,38]
*AUXIN RESPONSE FACTORS (ARF)*	Protein with N-terminal DNA binding domain, activator/repressor	Leaf polarity	*Arabidopsis*	[39]
*ANGUSTIFOLIA(AN3)/GRF-INTERACTING FACTOR1(GIF1)*	Transcription coactivators	Cell proliferation	*Arabidopsis*	[40,41]
*CLAVATA (CLV)*	CLV1 (receptor kinase); CLV2 (transmembrane protein); CLV3 (extracellular protein)	Maintain stem cell size	*Arabidopsis*	[42,43]
*Class-1 KNOTTED-like homeobox (KNOX1)*	Homeodomain protein	Maintain stem cell identity	*Arabidopsis*, *Zea mays*	[28,44]
*CUP-SHAPED COTYLEDON2 (CUC2)*	Protein containing the NAC DNA binding domain	Shoot meristem formation; organ boundary specification; and leaf margin development	*Arabidopsis*	[45]
*DEVELOPMENT-RELATE PcG TARGET IN THE APEX 4 (DPA4)*	RAV transcription repressor	Organ initiation and development; leaf margin development	*Arabidopsis*	[46]
*GROWTH-REGULATING FACTOR5 (GRF5)*	Transcription activators containing N-terminal QLQ or WRC domain	Cell proliferation	*Arabidopsis*	[47]
*Narrow sheath (ns)/PRESSED FLOWER (PRS)*	Homeodomain protein	Recruitment of leaf founder cells and leaf expansion/Marginal cell proliferation	*Zea mays Arabidopsis*	[44,48,49]
*PIN-FORMED1 (PIN1)*	Transmembrane protein	Auxin efflux	*Arabidopsis*	[34]
*PHANTASTICA (PHAN)*	MYB domain protein	Stem cell differentiation	*Antirrhinum majus*	[50]
*WUSCHEL (WUS)*	Homeodomain protein	Maintain shoot and floral meristem identity	*Arabidopsis*	[51]
*YABBY (YAB)*	Protein with zinc-finger and helix-loop-helix domains	Specification of Leaf polarity and lamina expansion	*Arabidopsis*	[52,53]

**Table 2 ijms-21-05132-t002:** MicroRNAs and their predicted targets in *Arabidopsis*.

miRNAs Family	Target Families/Gene	Biological Function	Species	References
miR156	SPL	Promote vegetative phase change and floral transition	*Arabidopsis*, *Zea mays*	[239,240,241]
miR159	MYB TFS:GAMYB, MYB33	Control floral identity and flower development	*Arabidopsis*	[242]
miR160	ARF	Leaf and root development, auxin response, floral organ identity	*Arabidopsis*	[229,243]
miR162	DCL1		*Arabidopsis*	[244]
miR164	NAC-TF: CUC1,CUC2	Shoot and root development	*Arabidopsis*, *Solanum*, and *Oryza*	[45,192,238,245]
miR164a	NAC-TF: CUC1, CUC2	Leaf development, patterning, and polarity		[26,246]
miR164c	NAC-TF: CUC1,CUC2	Floral identity and flower development		[247]
miR165/166	HD-ZIP, PHB	Meristem maintenance, vascular development and organ polarity	*Arabidopsis*	[160,248,249,250]
miR167	ARF6 and 8	Auxin response	*Arabidopsis*	[251]
miR168	AGO1		*Arabidopsis*	[252]
miR172	AP2	Developmental timing and floral organ identity	*Arabidopsis, Z. mays*, *S. tuberosum*	[253,254,255,256]
miR319	TCP	Leaf development	*Arabidopsis and Solanum lycopersicum*	[257]
miR390	TAS3	Auxin response, developmental timing, lateral organ polarity	*Arabidopsis*	[233,258]
miR393	F-box protein: TIR1	Hormone signaling for plant development	Arabidopsis, Oryza	[259]
miR396	GRF faimly	Control cell proliferation	*Arabidopsis*, *Medicago* and *Oryza*	[127,260,261]
miR408	Plantacyanin, Laccases	Stress response	*Arabidopsis*	[262]
TAS3	ARF3 and (only mosses) AP2 like	Leaf polarity	All land plants	[177,233]

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
