# Peer review of "Molecular and Hormonal Regulation of Leaf Morphogenesis in Arabidopsis"

_ijms, 2020, doi:10.3390/ijms21145132_

Round 1

Reviewer 1 Report

In this review Ali and Kahn describe in detail the existing data on the mechanisms controlling the development of Arabidopsis leaves. This topic is fundamental for the developmental biology, because it gives light to the mechanisms determining cell fate and organ shape. By date several studies have been performed on the determinants of leaf development, most of which are considered in the present review. However, this manuscript is too much descriptive and lacks a critical approach. Overall, the authors should try to reduce repetitions on the described mechanisms, and instead emphasize and comment the importance of some steps, making hypotheses etch…Also, in some point the reported genetic pathways are not precise or are mistaken. Moreover, as the discoveries done in Arabidopsis on leaf development have been crucial for our understanding on compound leaf development, I would suggest authors to briefly report these in a paragraph.

Minor points:

L122 STM induces IPT5 and IPT7 and not LOG4, please correct.

L142-153: this paragraph is the conceptual repetition of what it has been reported in previous paragraphs.

L269 in this line as in many mutants are reported in capitals, they should be reported in italic and lower case

L269 AS1 is an ARP gene (ASYMMETRIC LEAVES/ROUGHSHEET/PHANTASTICA), please correct.

In the manuscript are present several typos, please correct

In the manuscript gene names and acronyms are repeated in more than one place (e.g. L445)

L261 this point is not very clear. How SE regulate leaf development and hypothesis on this should be reported.

Author Response

Dear Editors and Reviewer

Thank you for giving me the opportunity to submit a revised draft of our manuscript titled Molecular and Hormonal Regulation of Leaf Morphogenesis in Arabidopsis. We appreciate the time and effort that you and the reviewers have dedicated to providing your valuable feedback on our manuscript. We are grateful to the reviewers for their insightful comments on our paper. We have been able to incorporate changes to reflect most of the suggestions provided by the reviewers. We have highlighted the changes within the manuscript.

Here is a point-by-point response to the reviewers' comments and concerns in bold font.

Comments and Suggestions for Authors

In this review Ali and Kahn describe in detail the existing data on the mechanisms controlling the development of Arabidopsis leaves. This topic is fundamental for the developmental biology, because it gives light to the mechanisms determining cell fate and organ shape. By date several studies have been performed on the determinants of leaf development, most of which are considered in the present review. However, this manuscript is too much descriptive and lacks a critical approach. Overall, the authors should try to reduce repetitions on the described mechanisms, and instead emphasize and comment the importance of some steps, making hypotheses etch…Also, in some point the reported genetic pathways are not precise or are mistaken. Moreover, as the discoveries done in Arabidopsis on leaf development have been crucial for our understanding on compound leaf development, I would suggest authors to briefly report these in a paragraph.

Response: Thank you so much for your comments and suggestions; we agree with these comments. We have read the whole manuscript and reduced the repetitions. Though the repetition was due to the involvement of almost similar genes in both the meristem maintenance and leaf initiation. As, this MS focused on the genetic networks involved in the maintenance and differentiation of meristematic tissues into a well-organized leaf shape; there are many genes that are involved in the meristematic tissue and also play a crucial role in leaf differentiation. We have added hypotheses and take that as a consideration to write this MS. Furthermore, according to your suggestion, the information that the understanding the basic knowledge of simple leaves (Arabidopsis thaliana) will help us to study the formation and genetic mechanisms involved in the compound leaves, were also added in the last section of introduction. 

Minor points:

L122 STM induces IPT5 and IPT7 and not LOG4, please correct.

Response: We apologize for the error found by the reviewer.                 We have revised this line.

L142-153: this paragraph is the conceptual repetition of what it has been reported in previous paragraphs.

Response: The correction has been made in the MS; we rearranged this paragraph.

L269 in this line as in many mutants are reported in capitals, they should be reported in italic and lower case

Response: We are thankful to the reviewer for this constructive comment. We checked the whole manuscript thoroughly, and italicized the genes/mutants, and the mentioned examples have also been corrected

L269 AS1 is an ARP gene (ASYMMETRIC LEAVES/ROUGHSHEET/PHANTASTICA), please correct.

Response: The correction has been made in the MS

In the manuscript are present several typos, please correct

Response: The MS has been thoroughly checked for typographical errors.

In the manuscript gene names and acronyms are repeated in more than one place (e.g. L445)

Response: We made this correction to the MS.

L261 this point is not very clear. How SE regulate leaf development and hypothesis on this should be reported.

Response: Based on this comment, we added more details and clarification about SE function by modulating meristem in leaf development.

Reviewer 2 Report

This manuscript entitled 'Molecular and hormonal regulation of leaf morphogenesis in arabidopsis' discuss how various factors previously studied in several studies modulate leaf development in the context of leaf initiation, polarity establishment, leaf flattening and shape. Authors describe not only the several fundamental aspects but also recent results that have sterngthened understanding of leaf development. This review on leaf development contributes to our overall comprehension of leaf biology, and this understanding can be used to improve crop production.

Author Response

This manuscript entitled 'Molecular and hormonal regulation of leaf morphogenesis in arabidopsis' discuss how various factors previously studied in several studies modulate leaf development in the context of leaf initiation, polarity establishment, leaf flattening and shape. Authors describe not only the several fundamental aspects but also recent results that have sterngthened understanding of leaf development. This review on leaf development contributes to our overall comprehension of leaf biology, and this understanding can be used to improve crop production.

Response: We appreciate the time and effort that the reviewer has dedicated for the review of this manuscript. We are grateful to the reviewers for recommending this MS for publication.

Round 2

Reviewer 1 Report

The authors answered to most of my concerns despite I would have prefer a more critical approach to the reported findings.

Minor comments:

The last paragraph in the introduction reporting the compound leaves has no reference.

Author Response

We would like to thank the Reviewer for his/her evaluation and for the constructive comments and suggestions that have helped us improve the quality of the manuscript. We have added the missing references in the MS as suggested by you.